# PLGA Based Nanospheres as a Potent Macrophage-Specific Drug Delivery System

**DOI:** 10.3390/nano11030749

**Published:** 2021-03-16

**Authors:** Barbora Boltnarova, Jana Kubackova, Josef Skoda, Alzbeta Stefela, Monika Smekalova, Petra Svacinova, Ivona Pavkova, Milan Dittrich, Daniel Scherman, Jarmila Zbytovska, Petr Pavek, Ondrej Holas

**Affiliations:** 1Department of Pharmaceutical Technology, Faculty of Pharmacy in Hradec Kralove, Charles University, Akademika Heyrovskeho 1203, 500 05 Hradec Kralove, Czech Republic; boltnarb@faf.cuni.cz (B.B.); kubackja@faf.cuni.cz (J.K.); smekalomo@faf.cuni.cz (M.S.); svacp3aa@faf.cuni.cz (P.S.); dittrich@faf.cuni.cz (M.D.); Jarmila.Zbytovska@vscht.cz (J.Z.); 2Department of Pharmacology and Toxicology, Faculty of Pharmacy in Hradec Kralove, Charles University, Akademika Heyrovskeho 1203, 500 05 Hradec Kralove, Czech Republic; skodajo@faf.cuni.cz (J.S.); horvata1@faf.cuni.cz (A.S.); pavek@faf.cuni.cz (P.P.); 3Department of Molecular Pathology and Biology, Faculty of Military Health Science, University of Defence in Brno, Trebesska 1575, 500 01 Hradec Kralove, Czech Republic; ivona.pavkova@unob.cz; 4Centre National de la Recherche Scientifique (CNRS), 3 rue Michel-Ange, 75016 Paris, France; daniel.scherman@parisdescartes.fr; 5Faculty of Chemical Technology, University of Chemistry and Technology, Technicka 5, 166 28 Prague, Czech Republic

**Keywords:** PLGA, nanospheres, nanoparticles, nanoprecipitation, inflammation, macrophages, drug delivery, drug targeting, liver, hepatic disease

## Abstract

Macrophages possess an innate ability to scavenge heterogenous objects from the systemic circulation and to regulate inflammatory diseases in various organs via cytokine production. That makes them attractive targets for nanomedicine-based therapeutic approaches to inflammatory diseases. In the present study, we have prepared several different poly(lactic-co-glycolic acid) (PLGA) polymer nanospheres for macrophage-targeted drug delivery using both nanoprecipitation and emulsification solvent evaporation methods. Two experimental linear PLGA polymers with relatively low molar weight, one experimental branched PLGA with unique star-like molecular architecture, and a commercially available PLGA, were used for nanosphere formulation and compared to their macrophage uptake capacity. The nanosphere formulations labelled with loaded fluorescent dye Rhodamine B were further tested in mouse bone marrow-derived macrophages and in hepatocyte cell lines AML-12, HepG2. We found that nanospheres larger than 100 nm prepared using nanoprecipitation significantly enhanced distribution of fluorescent dye selectively into macrophages. No effects of nanospheres on cellular viability were observed. Additionally, no significant proinflammatory effect after macrophage exposure to nanospheres was detected as assessed by a determination of proinflammatory cytokines *Il-1β* and *Tnfα* mRNA. All experimental PLGA nanoformulations surpassed the nanospheres obtained with the commercially available polymer taken as a control in their capacity as macrophage-specific carriers.

## 1. Introduction

Macrophages together with monocytes and dendritic cells form a mononuclear phagocyte system (MPS), which plays an important role in maintaining homeostasis. The MPS is disseminated through various tissues as well as in the blood stream. The main functions of MPS cells are blood debris clearance and protection from various pathogens, both occurring through phagocytosis [1]. Liver resident macrophages, the Kupffer cells, are the most numerous cell population of MPS. They are among the key factors influencing the course of various hepatic diseases [2,3].

In response to microenvironmental stimuli and signals (e.g., corticosteroids, lipopolysaccharide and interleukins) [4,5], macrophages undergo polarization into several subtypes with specific phenotype and biological functions. There are two main subtypes: M1 and M2, and this macrophage polarization plays an important role in the development of inflammation as well as in antitumor immunity. Activated M1 macrophages display a pro-inflammatory activity, which has a beneficial protective effect against pathogens as well as cancer development by activating tumour killing mechanisms, but which can lead to the detrimental development of chronic inflammation. M2 activated macrophages, also termed alternatively activated, mediate anti-inflammatory processes such as wound healing and tissue remodelling. On the other hand, they suppress adaptive tumour-specific immune response and promote tumour growth, invasion and metastasis [6,7].

This switch of macrophage polarization can potentially be therapeutically utilized, but the challenging point is to deliver compounds with the capacity to influence polarization directly to macrophages, in order to avoid the unwanted systemic effects of pleiotropic drugs such as those of corticosteroids [8]. A promising approach is to load active compounds in an appropriate nanoformulation. This regulation of macrophage polarization via a suitable nanocarrier is an interesting therapeutic strategy to manage a wide variety of diseases, including chronic inflammation, tumour growth, or immune system disorders [6].

A variety of nanoparticle types are used for drug delivery, among them liposomes, nanocapsules and nanospheres. Liposomes are composed of a mono or multiple lipid bilayers. Nanocapsules comprise a polymeric shell delimitating an inner drug-loaded space. Nanospheres possess a polymeric matrix containing the drug of interest, which must display a predominantly hydrophobic character for optimal nanosphere loading. For specific drug delivery to macrophages, the currently proposed nanocarriers are almost exclusively limited to liposomes and lipid particles. Liposomes and lipid particles are convenient drug delivery systems; however, they suffer from certain disadvantages, such as problematic long-term stability, drug leakage during storage and non-specific distribution [9,10].

Drug-loaded solid polymeric nanoparticles called nanospheres represent a promising tool for the selective delivery of an active drug to macrophages, since hydrophobic particles with the size range of 100–300 nm are very effectively scavenged by macrophages from the systemic circulation within few minutes after intravenous administration due to high concentration of phagocytic cells in the liver and, to the lesser extent, in the spleen [11]. Other desirable properties of a suitable nanoformulation are biodegradability, non-toxicity, sufficient drug loading efficiency, and the possibility of surface modification [12].

The material composed of poly (lactic-co-glycolic) acid (PLGA) fulfils all these requirements. PLGA is a hydrophobic polymer widely used for nanoparticle and nanosphere formulation. The FDA has classified PLGA as generally recognized as safe (GRAS). The main degradation products are lactic and glycolic acid, which are readily eliminated from the body via the citric acid cycle [13,14]. The physico-chemical properties of PLGA are determined by lactic to glycolic acid ratio, molar mass and molecular architecture. PLGA is a well-established material for design of controlled release drug delivery systems. Pharmaceutics, regardless of their molar weight, can be loaded into PLGA drug delivery systems and consequently released from the matrix in a controlled manner. Consequently, a wide variety of active compounds can be used as payload of PLGA based drug delivery systems ranging from small molecules such as anti-tumour drugs with molar weight 0.5 kDa [15] to proteins and enzymes with molar mass up to 50 kDa [16,17]. The drug release profile of PLGA based drug delivery systems is usually characterized by the initial burst of a loaded drug followed by zero order kinetics release [18]. The drug release profile is influenced by PLGA degradation rate, initial molar mass, fabrication method or pH of environment and the nature of PLGA-drug interactions [19].

The PLGA that is usually used for medical applications has a molecular weight of several tens of thousands of Daltons. Such molecular weight is not necessarily the most advantageous for nano drug delivery systems purposes. Therefore, we present PLGA nanospheres fabricated from several structural variants of PLGA with molecular weights ranging between 3500 Da and 14,000 as well as commercially available PLGA with molecular weight of 17,000 Da. The PLGA polymers varying in molar weight, lactic to glycolic acid ratio and molecular architecture used in the study have been previously described [20].

Using the fluorescent dye Rhodamine B (RhB) as a loaded compound to make nanospheres traceable, the PLGA nanospheres were evaluated in terms of cellular uptake by bone marrow-derived macrophages (BMM), mouse AML-12 normal hepatocytes, and the HepG2 immortalized human hepatic cell line. We presently report that nanospheres formulated using the nanoprecipitation method possess satisfactory drug loading efficiency, are non-toxic, and are preferably engulfed by macrophages without inducing production of pro-inflammatory markers. Therefore, these PLGA nanospheres are suitable candidates for development of a potent macrophage-specific drug-delivery system.

## 2. Materials and Methods

### 2.1. Materials

The non-commercial experimental PLGA polymers were synthesized and characterized as described previously [20]. Briefly, linear copolymer of PLGA was synthesized by hot-melt condensation polymerization from equimolecular mixture of glycolic acid (GA) and DL-lactic acid (LA). For branched copolymer, GA and LA in equimolar ratio together with 2% polyacrylic acid were used. All the chemicals were obtained from Merck (Darmstadt, Germany). The synthesis of linear and branched PLGA with lower molar mass was carried out at 160 °C and 550 Pa for 75 h. The reaction period of branched PLGA with higher molar mass lasted for 90 h.

Experimental polymers PLGA 50:50 (equimolar LA:GA ratio, M_w_: 2400 g/mol), PLGA 70:30 (7:3 LA:GA ratio, M_w_: 3200 g/mol), polyacrylic acid branched PLGA (2% polyacrylic acid, equimolar LA:GA ratio, M_w_: 14,400 g/mol) were synthesized. The commercially available PURASORB^®^ 5002 (equimolar LA:GA ratio, M_w_: 17,000 g/mol, Corbion, Amsterdam, The Netherlands) was used as the standard control material. Rhodamine B (RhB), Pluronic^®^ F-127, ethyl acetate, and acetone were purchased from Merck (Darmstadt, Germany).

### 2.2. Nanosphere Preparation

The nanoprecipitation method (NPM) was the first method used for nanosphere preparation [21]. Briefly, 30 mg PLGA and 250 µg RhB were dissolved in 1 mL of acetone and injected into 10 mL of 0.1% (*w/v*) aqueous solution of Pluronic^®^ F127 under constant magnetic stirring (300 rpm). The mixture was stirred for two hours in order to ensure complete acetone evaporation.

The emulsification solvent evaporation method (ESE) was the second method used in the study [22]. Briefly, 30 mg PLGA and 150 µg RhB were dissolved in 1 mL of ethyl acetate to create the organic phase. 10 mL of 0.5% (*w/v*) Pluronic F127 aqueous solution was used as the aqueous phase. The organic phase was added to 5 mL of the aqueous phase and subjected to high energy sonication using the ultrasonic probe (MS 1.5 Mikrospitze, Bandelin, Germany) for 1 min at 80% power output while being cooled by ice bath. After sonication, the created emulsion was added to further 5 mL of the aqueous phase, followed by two hours of solvent evaporation at atmospheric pressure and gentle stirring.

Raw nanosuspension prepared either by NPM or ESE was centrifugated at 10,000× *g* at 12 °C for 15 min. The supernatant containing stabilizer, non-loaded RhB and PLGA debris was removed and pellet was dispersed in 10 mL of deionized (DI) water. Collected nanospheres were filtered through a 0.45 μm filter (Minisart, celulose acetate membrane, 28 mm, Sartorius, Göttingeng, Germany). Two more centrifugation cycles with the aforementioned settings followed, and finally the pelleted nanospheres were re-dispersed under aseptic conditions using sterilized DI water.

### 2.3. Nanosphere Characterization

Prepared nanospheres were characterized in terms of their size and polydispersity using dynamic light scattering (Zetasizer Nano ZS, Malvern Panalytical, Malvern, UK). Samples were diluted 20 times by DI water. The intensity of the scattered light was detected at a backscattering angle of 173°. The size and polydispersity of nanospheres were calculated based on the intensity size distribution protocol. This setup was chosen to avoid results biased by varying optical properties of loaded and non-loaded nanospheres. Viscosity of media was selected to that of the DI water. All samples were measured three times and each measurement involved 12 independent runs. Zeta potential was determined using electrophoretic light scattering (Zetasizer Nano ZS, Malvern Panalytical) For zeta potential measurement, attenuator was set to automatic mode. DTS1070 folded capillary cuvettes were used.

Drug loading efficiency (DLE) was determined according to the following protocol. Purified nanospheres were centrifugated at 20,000× *g* for 20 min. Supernatant was removed and pellet of nanospheres was dissolved in acetonitrile. The concentration of RhB incorporated in the nanospheres was determined spectrophotometrically by absorbance at 555 nm (Specord 250 Plus, Jena Analytik, Jena, Germany). DLE measurement was performed at the same time point as cells treatment (as described in Section 2.4.2) to evaluate the amount of RhB loaded into nanospheres that was used for cell uptake assay. The following equation was used to calculate RhB loading efficiency:(1)$$\%DLE=\frac{RhB\;loaded}{RhB\;used\;for\;preparation}$$

Prepared nanospheres were visualised using atomic force microscopy (AFM) using the Nanosurf easyScan 2 FlexAFM instrument (Nanosurf, Switzerland). AFM topological images were prepared to verify spherical shape and size uniformity of prepared nanospheres. Samples were prepared by dilution with DI water at the ratio 1:50. A drop of this dispersion was applied to a microscope slide and dried at room temperature. Scans were performed in Tapping Mode using the cantilever with nominal spring constant of 48 N/m and resonant frequency of 190 kHz. Resolution was 512 × 512 points.

### 2.4. Cells Isolation, Differentiation and Exposure

The bone marrow-derived macrophages (BMM) were derived from 6–10-week-old BALB/c female mice (Velaz, Czech Republic), as described by Weischenfeldt and Porse [23] with minor modifications. Briefly, the bone marrow cells were flushed out from dissected femurs and tibias. The obtained cells were differentiated into macrophages in 100 mm bacterial Petri dishes in Dulbecco’s Modified Eagle Medium (DMEM, HyClone) supplemented with the heat inactivated 10% (*v/v*) fetal bovine serum (FBS, USA origin from Merck), 20% L929-conditioned medium (source of macrophage-colony stimulating factor), and antibiotics—50 µg/mL streptomycin and 50 U/mL penicillin (for the first three days of cultivation only) at 37 °C and 5% CO_2_. At day 7, the differentiated cells were detached by incubation in ice-cold Phosphate-Buffered Saline (PBS) (4 °C, up to 10 min) followed by gentle pipetting. Pelleted cells were resuspended in fresh DMEM with 10% FBS and seeded onto 48 or 96 well polystyrene plates in density 250,000 cells/cm^2^. For M1 polarization, the BMMs were treated with *Escherichia coli* lipopolysaccharide (LPS; Merck, Darmastadt, Germany) in a concentration of 100 ng/mL for 2 h [24].

The mouse hepatocyte AML-12 cells (ATCC^®^ CRL-2254™) were cultured in DMEM/F12 medium supplemented with 10% FBS, 10 μg/mL insulin, 5.5 μg/mL transferrin, 5 ng/mL selenium and 40 ng/mL dexamethasone (all components from Merck). The AML-12 cell line was established from hepatocytes from a mouse (CD1 strain, line MT42) transgenic for human TGFα. The cells were seeded onto 48 or 96 well plates (Nunc^TM^) in a density of 42,000 cells/cm^2^.

Human hepatocyte tumour-derived cells (HepG2) obtained from the European Collection of Authenticated Cell Cultures, Salisbury, UK, were maintained in DMEM (Invitrogen, Carlsbad, CA, USA) supplemented with 10% (*v/v*) FBS, 2 mM glutamine and 1 mM sodium pyruvate (all components purchased from Merck). The cells were seeded onto 48 or 96 well plates in a density of 42,000 cells/cm^2^.

#### 2.4.1. Cellular Viability Assay

The MTS assay–CellTiter 96^®^ Aqueous One Solution Cell Proliferation Assay (Promega, Hercules, CA, USA) was used according to the manufacturer’s protocol for viability testing in all cellular models. Briefly, 24 h after cell seeding, nanospheres were added. After two washes with PBS, cells were incubated with the CellTiter reagent for 1 h at 37 °C. After that, absorbance at 490 nm was measured and cell viability was determined relative to vehicle (sterile water)-treated cells (100% viability). A 10% (*w/v*) sodium dodecyl sulphate (SDS) solution was used as the cytotoxic control. The experiments were repeated at least three times, and the tested formulations were evaluated in triplicates in all experiments. The threshold of 80% viability was used as the limit for potential cytotoxicity.

#### 2.4.2. Nanosphere Cell Uptake Quantification

Nanospheres were prepared immediately before the cell treatment to keep the RhB leakage to the minimum (Appendix A). Nanospheres were added to cells 24 h after seeding. Nanospheres were diluted in the cell culture media in a concentration of 300 μg/mL PLGA nanospheres. This concentration was lower compared to previous studies [25] in order to avoid non-facilitated cell uptake. As a control, the cells were incubated with 15 µg/mL free RhB dissolved in media. The cells were incubated in the presence of RhB labelled nanospheres or free RhB for 1 and 4 h. Subsequently, the culture medium was removed, and the cells were washed twice by PBS. A volume of 150 µL PBS per well was added, and the cells were frozen overnight. After thawing, the measurement of nanospheres loaded RhB fluorescence was performed using the Synergy 2 Biotek plate reader (BioTek, Winooski, VT, USA) with filters for excitation 495/10 nm and emission 590/35 nm, respectively. The cellular uptake of RhB-labelled nanopheres or free RhB was quantified in arbitrary units using the following equation where F stands for fluorescence value and %DLE stands for the efficiency of RhB incorporation into nanospheres:(2)$$cell\;uptake=\frac{100\times F}{\%DLE}$$

Experiments were performed at least three times. The data are presented as means with standard deviation (SD).

#### 2.4.3. Real-Time Quantitative PCR (RT-qPCR)

RNA isolation was performed using TRI Reagent^®^ (Merck, Darmastadt, Germany) according to the manufacturer’s protocol. EconoSpin^®^ columns (Epoch Life Science, Missouri City, TX, USA) were used for purification. The purity and the concentration of RNA was measured by NanoDrop spectrophotometer. For the transcription, RevertAid RT Kit (Thermo Fisher Scientific, Waltham, MA, USA) was used. The qRT- PCR experiments were performed using the QuantStudio 6 Real-Time PCR System with TaqMan Fast Advanced Master mix (ThermoFisher Scientific, Waltham, MA, USA). Pro-inflammatory activity in BMM was evaluated by the detection of two cytokines *Il-1β* and *Tnfα* mRNA using commercial TaqMan assays Mm00434228_m1 for *Il-1β* and Mm00443258_m1 for *Tnfα* (ThermoFisher Scientific, Waltham, MA, USA). The house-keeping hypoxanthine-guanine phosphoribosyltransferase *Hprt* gene was used as an internal standard (Mm 03024075_m1) (ThermoFisher Scientific, Waltham, MA, USA). PCR reactions were performed using technical replicates. The delta-delta method was used for relative mRNA expression quantification.

### 2.5. Confocal Microscopy

For laser scattering confocal microscopy, BMM were seeded onto confocal dishes (SPL, glass bottom, 101350, SPL Life Sciences, Gyeonggi-do, Korea) in a density of 100,000 cells/cm^2^. After overnight stabilization, cells were treated for 1 h with vehicle (DMEM), RhB solution (15 µg/mL) or with RhB-loaded nanosphere samples (300 μg/mL). Living cell nuclei were stained with Hoechst 33,342 (0.2 µM, 5 min at 37 °C). For microscopy Nicon Ti ECLIPSE microscope and Nikon A1 plus camera (Nikon Instruments, Melville, NY, USA) using 405 and 561 nm lasers was used. The pinhole diameter was set to 19.16 µM and microphotographs were taken using the NIS Elements AR 4.20 software (Laboratory Imaging, Prague, Czech Republic). Three representative photographs of every treatment were taken.

### 2.6. Statistics

The Prism 8 program (GraphPad, San Diego, CA, USA) was used for the statistical analysis of cell uptake quantification, RT-qPCR, and cytotoxicity assays. The statistical significance of differences between the means of individual groups was calculated using a one-way analysis of variance (ANOVA) with Dunnett’s post hoc test. In addition, we used a student’s unpaired t-test for the comparison of two groups of values. A *p*-value of *p* < 0.05 was considered to be statistically significant.

## 3. Results and Discussion

### 3.1. Characteristics of PLGA Nanospheres

The granulometry and zeta potential of nanospheres prepared from polymers PLGA 50:50, PLGA 70:30, PLGA branched by polyacrylic acid and Purasorb^®^ are summarized in Table 1. The nanospheres prepared by NPM were generally larger (130–170 nm) relative to nanospheres obtained by ESE (smaller than 100 nm). The size of nanospheres prepared by ESE was governed by Pluronic F127 concentration. Minimum threshold size of droplets formed during the organic phase emulsification was lower with higher Pluronic F127 concentration due to the steric stabilization effect as well as a decrease in surface tension. This was the main reason for using a higher concentration of Pluronic F127 for ESE method as compared to NPM (0.5% and 0.1%, respectively). On the other hand, NPM prepared nanospheres were formed rapidly during the initial stage of solvent mixing, and their size was less susceptible to dependence on the Pluronic F127 concentration [26].

The polydispersity index (PDI) of all prepared nanospheres was lower than 0.2, indicating sufficient monodispersity (Figure 1A). This was verified using AFM imaging (Figure 1B), where the regular spherical shape of nanospheres was also clearly visible. The zeta potential of prepared nanospheres ranged from −22 mV to −33 mV, suggesting a good colloidal stability. Moreover, interactions between nanomaterials and MPS cells were significantly determined by nanomaterials’ surface charge [27]. In terms of macrophage specific delivery, a negative charge is the most appropriate characteristic and carboxyl functionalized negatively charged nanoparticles promote uptake by macrophages as compared to non-differentiated monocytes [28]. This was confirmed by our results showing that nanospheres prepared from branched polymer, rich in carboxyl groups, were better suited for macrophage-specific delivery owing to their more pronounced negative surface charge (see results below).

### 3.2. Nanosphere Loading Efficiency

The results of RhB loading efficiency are summarized in Table 2. Our data indicate that the NPM is a more suitable method for the incorporation of RhB into PLGA nanospheres, as compared to ESE. The emulsification solvent evaporation method has been reported to be susceptible to leakage of loaded compounds at the phase of primary emulsion [29]. We observed that RhB loading efficiency was PLGA dependent since RhB as a water-soluble molecule bearing a positive charge within its structure incorporated itself into the PLGA nanosphere in higher quantity into lower molar weight PLGA (Samples **1b**, **2b**) or branched PLGA (**3b**). This suggests a role of ion pairing mechanism in incorporation of RhB as these polymers are characterized by higher concentration of terminal carboxyl groups within the material compared to the same amount of higher molar weight polymeric material such as Purasorb^®^ grade used as a standard control material. The highest loading efficiency was achieved using NPM and branched polymer (formulation **3b**), which are characterized by high carboxyl content.

### 3.3. Cellular Uptake of Nanospheres

#### 3.3.1. Internalization

Cellular uptake of RhB loaded nanospheres prepared in this study was evaluated in BMM and in hepatocyte-derived cell lines HepG2 (human) and AML-12 (mouse) The BMM were chosen as a critical part of MPS and as important immune cells regulating inflammatory processes in the liver. AML-12 cell line was selected as a model hepatocyte cell line with no phagocytic activity to distinguish macrophage-specific targeting to assess potential side effects due to off-target delivery. Nanospheres for the cell uptake experiments were prepared immediately before the experiment and used directly for cell treatment. The results showed that released RhB (see Appendix A) is not actively taken up by any of tested cell lines, so the fluorescence presented in Figure 2 is mediated by RhB loaded within nanospheres upon cell entry. Our results presented in Figure 2A show that BMM actively engulfed the nanospheres at a varying rate and that this phenomenon was size dependent. Noticeably, nanospheres prepared by ESE with the size below 100 nm were not actively engulfed by BMM, irrespective of the kind of PLGA used for their fabrication (samples **1d**, **2d**, **3d**). However, PLGA nanospheres prepared by NPM, which are characterized with the size of around 150 nm, were internalized at a significantly higher rate. This was to be expected, as the nanoparticle size is one of the determining factors for macrophage targeting [30]. Nanospheres of the size surpassing 100 nm have been repeatedly reported as suitable for in vivo administration as well as capable of accumulation in macrophages [31,32].

Moreover, it is noteworthy that, nanosphere uptake by BMMs was largely superior to the uptake by HepG2 or AML-12 cells (Figure 2A) clearly indicating selective targeting into phagocytic cells such as BMMs.

In the case of nanospheres prepared within this study, PLGA type used for nanospheres preparation also determines the internalization rate. All samples prepared by the NPM including commercially available PLGA (Purasorb^®^) exceeded the internalization rate of RhB in solution (Figure 2A, first columns) confirming suitability of PLGA nanospheres for macrophage targeting. Linear PLGA 70:30, and the branched polymer of star-like architecture yielded nanospheres with the highest internalization rate by BMM in this study (**2b**, **3b**) surpassing both commercially available PLGA and free RhB in solution. PLGA 70:30 is characterized by increased lipophilicity owing to higher lactic acid ratio and showed the best internalization rate among the tested PLGA-based materials. These findings correlate well with previously published data suggesting that lipophilic objects are recognized and scavenged by macrophages more frequently compared to the hydrophilic ones. Opsonization plays a crucial role in this process in vivo, as hydrophobic particles are readily opsonized and thereby made easily recognizable for phagocytic cells [31]. Our data show that the hydrophobicity of nanospheres itself enhances internalization in cellular models even without opsonization.

The branched and low molar weight PLGA is characterized by a higher number of terminal OH and COOH groups. Formulations **2b** and **3b** with zeta potential of −29 mV and −33 mV respectively showed a significantly higher internalization rate (*p* < 0.05) (Figure 2A) compared to Purasorb-based formulation **4b** (zeta potential −24 mV) confirming that surface charge is one of the determining factors for macrophage specific drug delivery. This can be observed despite the possible mutual electrostatic repulsive interactions between the cell membrane and negatively-charged nanospheres. Macrophages, being professional phagocytes, engulfed negatively charged particles under in vitro conditions. This is in agreement with previously published data suggesting that negatively-charged particles are subjected to active CD 64 mediated phagocytosis in the presence of serum proteins and by clathrin and dynamin endocytosis when protein corona is not or cannot be formed [33,34,35].

The internalization of PLGA nanospheres is a time-dependent phenomenon. In agreement, the amount of nanospheres accumulated within the first hour after the exposure was significantly lower compared to the amount accumulated after 4 h irrespective of LPS treatment. We observed that more than 50% of nanospheres were internalized within the first hour for linear polymers. This is in contrast to the data published for macrophage targeted liposomes. The amount of internalized liposomal formulation after 1 h was insignificant under similar conditions [36]. Thus, PLGA nanospheres provide a tool for specific and rapid macrophage drug delivery. The somehow slower cell entry rate of branched PLGA nanospheres was observed within the first hour. The amount of nanospheres internalized after 1 h was lower compared to the linear PLGA counterparts (**3b** compared to **1b** and **2b**). Cellular uptake by LPS-stimulated macrophages and non-stimulated macrophages was also investigated. Notably, proinflammatory LPS-treated macrophages seemed to be more active in terms of their scavenging activity. The amount of PLGA nanospheres **2b** and **3b** engulfed by the polarized macrophages was significantly higher compared to the non-polarized ones at both time points (i.e., in the absence of LPS) (Figure 2A).

On the other hand, our data shown in Figure 2B,C demonstrate that the rate of internalization of tested nanospheres into hepatocytic cell lines AML-12 and HepG2 was negligible and significantly lower compared to RhB solution. Formulation **3b** was the only formulation enabling delivery of RhB into hepatic cells in a comparable degree to RhB solution. There was no significant relationship between either type of PLGA used or the nanosphere size and the cell uptake by AML-12 or HepG2 cells. This suggests that PLGA nanospheres do not act as hepatocyte-specific drug delivery system and do not enter hepatocyte cells which are devoid of phagocytic activity. Importantly, it is desirable to avoid the distribution of macrophage-targeted nanospheres into hepatocytes. Indeed, numerous active compounds (e.g., nuclear receptor ligands, such as corticosteroids) intended or used for treatment of hepatic inflammatory disorders or steatohepatitis show side effects via triggering undesirable metabolic pathways in hepatocytes. This may result in hyperlipidaemia or diabetes, upon distribution of anti-inflammatory pharmaceutics into hepatocytes. [37,38]. A previously published study shows preferential distribution of PLGA nanoparticles (>250 nm) to Kupffer cells to a lesser extent to the liver sinusoidal endothelial cells accompanied with negligible distribution into hepatic stellate cells and hepatocytes. In the liver, the size of the fenestrae among the liver sinusoidal endothelial cells is estimated to be about 280 nm [39]. Therefore, nanoparticles size should be smaller than 280 nm for nanoparticles to avoid being trapped by these scavenging cells. These data indicate that nanoparticles prepared in this study of about 160 nm in size should avoid distribution into liver sinusoidal cells and act as a macrophage specific drug delivery system.

#### 3.3.2. Confocal Microscopy

Confocal microscopy was used to verify the internalization rate and cellular localization of our nanospheres into BMMs. Our PLGA samples were compared with control untreated cells and cells treated by free RhB solution. Representative confocal microscopy image of formulation **3b** shows that the amount of engulfed nanospheres was considerable, and that a much higher quantity of RhB was taken up compared to RhB solution in media (Figure 3). PLGA nanospheres **3b** were located intracellularly and did not adhere to the outer cell membrane.

### 3.4. Inflammatory Effect

RT-qPCR was chosen as a method for cytokines level evaluation as an approach widely accepted in analysis of immune activation in macrophages [40,41].

RT-qPCR data show (Figure 4) that RhB-loaded nanospheres did not induce proinflammatory polarization of macrophages and no drastic increase in proinflammatory cytokines production increase after engulfment was observed. The significant increase of *Tnf*α mRNA after exposure to sample **3b** (Figure 4A) might have resulted from some mild proinflammatory activity of branched polymers versus linear ones. However, the proinflammatory behaviour of the **3b** nanospheres was not confirmed by *Il-1*β mRNA expression (Figure 4B). Interestingly, RhB solution itself stimulated up-regulation of *Il-1β* mRNA in BMM cells. LPS significantly upregulated, but dexamethasone significantly suppressed expression of both *Tnfα* and *Il-1β* mRNA, thus validating the assay. This data suggests a benefit of PLGA nanospheres compared to lipid based nanoformulations, since a previously published study by Bartneck et al. demonstrated proinflammatory effects of dexamethasone loaded liposomes in non-polarized macrophages [36].

### 3.5. Effect of Nanospheres on Cellular Viability

Effects of PLGA nanospheres on cellular viability was tested for all cell lines including mouse BMMs. A viability of 80% is generally accepted as the threshold value for significant toxicity. None of the formulations evaluated in our experiments caused a drop in viability below this value. We showed (Figure 5A) that even after the substantial engulfment by macrophages, there was no statistically significant decrease in cellular viability in BMM caused by PLGA nanospheres. Data shown in Figure 5B suggest that PLGA nanospheres were also well tolerated by HepG2 cells, and no toxic effect was observed. AML-12 cells are generally more sensitive compared to HepG2. However, there was no significant toxicity to be seen in AML-12 cells either (Figure 5C).

## 4. Conclusions

Four structurally-different PLGA variants were used to prepare fluorescently labelled nanospheres. Two distinct methods of preparation were employed, the nanoprecipitation method and the emulsion solvent evaporation method. Resulting nanospheres were spherical and monodisperse, as well as sufficiently loaded with the fluorescent tracer RhB. Size, lipophilicity and surface charge of the assessed nanospheres were shown to be determining factors for drug loaded PLGA nanosphere internalization into macrophages. Nanospheres smaller than 100 nm were negligibly phagocyted by mouse bone marrow-derived macrophages, regardless of other parameters. PLGA lactic and glycolic acid ratio used for nanosphere fabrication affected BMM cell uptake under in vitro conditions. A more pronounced surface charge promoted phagocytosis. Therefore, PLGA with more profound surface charge and/or higher lactic/glycolic acid ratio are considerably more convenient macrophage-specific drug delivery systems. Nanospheres based on PLGA with low molar weight or with branched molecular architecture can serve as a highly potent, biodegradable and biocompatible drug delivery platform for macrophages.

Importantly, PLGA nanosphere uptake by BMMs is largely superior to uptake by hepatocytes, and the M1-macrophages polarized by lipopolysaccharide seem to possess higher phagocytic activity for solid nanospheres. Thus, these findings strongly suggest that PLGA-based nanospheres of optimized characteristics represent a promising therapeutic tool to treat liver inflammatory disease by their availability to target selectively activated Kupffer cells in the liver. The liver inflammatory diseases represent a growing global health concern, which is caused by excess fatty food intake, and is reflected by the spreading of non-alcoholic steatohepatitis (NASH), which can ultimately lead to hepatocarcinoma.

Finally, from an industrial and scale-up point of view, nanospheres prepared by the simple nanoprecipitation method are taken-up by mouse BMMs much more efficiently owing to their larger size compared to the nanospheres prepared by the multi-step emulsification solvent evaporation method.

## Figures and Tables

**Figure 1 nanomaterials-11-00749-f001:**
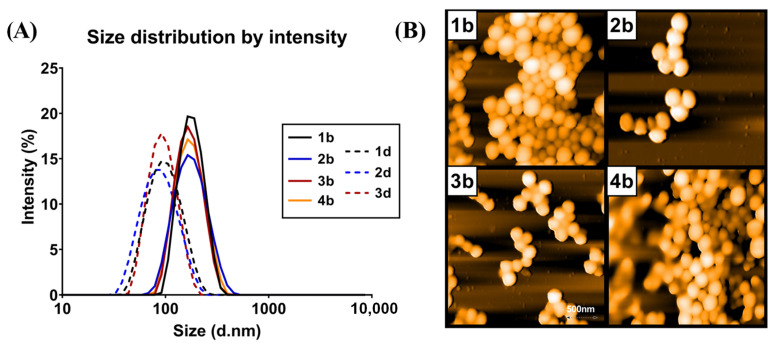
Granulometric characteristics of PLGA nanosphesres: (**A**) Size distribution by scattered laser light intensity of RhB loaded nanospheres; (**B**) atomic force microscopy (AFM) topographic images of samples (**1b**), (**2b**), (**3b**) and (**4b**) as described in Table 1. AFM was performed using the Nanosurf easyScan 2 FlexAFM instrument with samples diluted at the ratio 1:50 on a dried microscope slide.

**Figure 2 nanomaterials-11-00749-f002:**
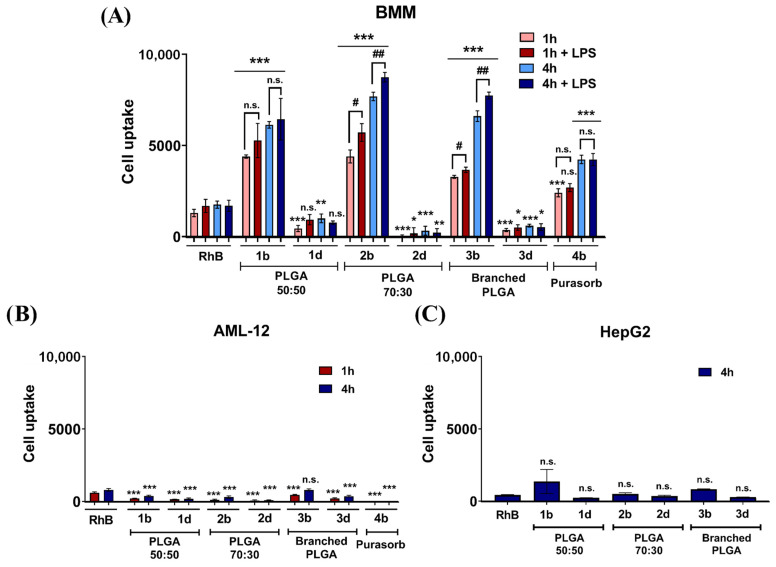
Accumulation of nanospheres in (**A**) Bone marrow-derived macrophages (BMM) either without stimulation or stimulated by lipopolysaccharide (LPS); (**B**) AML-12 cells; and (**C**) HepG2 cells. The accumulation was assessed by measuring the fluorescence of nanosphere-incorporated Rhodamine B (RhB) after 1 and 4 h of incubation. The terms 1b, 1d, etc. refer to the formulations presented in Table 1 and Table 2 and refer to the use of the nanoprecipitation method (NPM) for the b-indexed formulations, and of the emulsification solvent evaporation method (ESE) for the d-indexed formulations, respectively. The values were weighted taking into account the loading efficiency of the individual formulations. The results are compared to a solution of free RhB in concentration 15 µg/mL. * *p* < 0.05, ** *p* < 0.01, *** *p* < 0.001, statistically significant difference of cell entry rate in comparison to RhB solution; ^#^
*p* < 0.05, ^##^
*p* < 0.015 statistically significant effect of BMM stimulation on a nanosphere cell uptake.

**Figure 3 nanomaterials-11-00749-f003:**
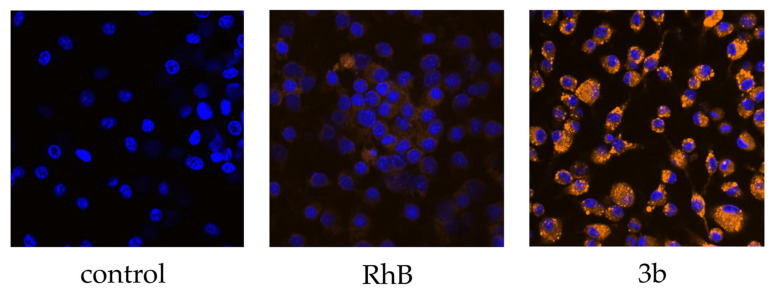
Confocal microscope imaging of mouse bone marrow-derived macrophages untreated (control), treated with solution of free Rhodamine B in concentration 15 µg/mL (RhB) and with formulation **3b** representing nanospheres fabricated from branched PLGA using NPM. Cells were incubated with the samples for 1 h. The nuclei were stained by dye Hoechst 33342. Magnification 40× was used.

**Figure 4 nanomaterials-11-00749-f004:**
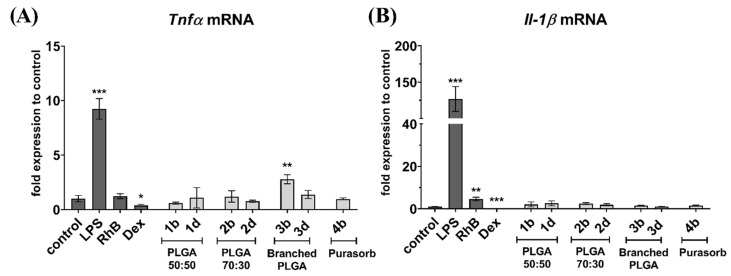
Real-time RT-qPCR quantification of (**A**) Tnfα mRNA and (**B**) Il-1β mRNA in bone marrow-derived macrophages. Cells were incubated with nanospheres for 24 h. Results were compared with untreated cells (control), cells treated with lipopolysaccharide in concentration 100 ng/mL (LPS) to stimulate proinflammatory cytokines expression, dexamethasone solution in a concentration of 200 nM/mL (DEX) and RhB solution in concentration 15 µg/mL (RhB). Data are presented as a fold change in expression relative to untreated control cells (set to be 1). Figure shows a representative experiment data. * *p* < 0.05, ** *p* < 0.01, *** *p* < 0.001.

**Figure 5 nanomaterials-11-00749-f005:**
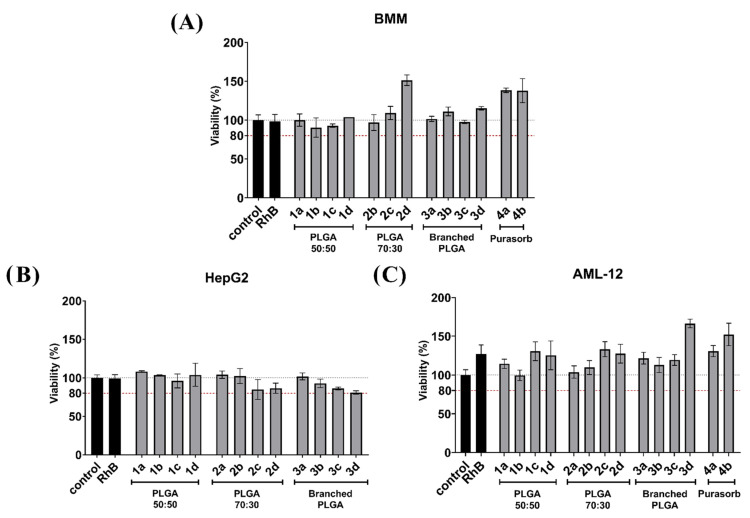
Viability of (**A**) bone marrow-derived macrophages (BMM), (**B**) HepG2, or (**C**) AML-12 cells after treatment with nanospheres. Cells were exposed to nanospheres for 24 h in the concentration of 300 μg/mL and viability was evaluated by the MTS assay. Viability in cells treated with nanospheres was compared to untreated cells (control). Solution of Rhodamine B in a concentration of 15 µg/mL (RhB) was used as the control for the model substance. A threshold of 80% viability was set as a threshold value for toxic effect.

**Table 1 nanomaterials-11-00749-t001:** Characteristics of PLGA nanospheres used in the study prepared by the nanoprecipitation (NPM) and the emulsification solvent evaporation (ESE) methods.

Batch No.	Polymer	Preparation Method	Fluorescent Tracer	Size (nm) ± SD	PDI ± SD	Zeta Potential (mV) ± SD
1a	PLGA 50:50	NPM	-	157.5 ± 3.7	0.06 ± 0.02	−27 ± 3
1b	PLGA 50:50	NPM	RhB	171.5 ± 1.4	0.10 ± 0.01	−25 ± 2
1c	PLGA 50:50	ESE	-	91.1 ± 1.2	0.09 ± 0.01	−22 ± 3
1d	PLGA 50:50	ESE	RhB	91.2 ± 5.1	0.16 ± 0.03	−22 ± 1
2a	PLGA 70:30	NPM	-	138.0 ± 6.5	0.07 ± 0.03	−28 ± 2
2b	PLGA 70:30	NPM	RhB	166.7 ± 1.6	0.08 ± 0.01	−29 ± 3
2c	PLGA 70:30	ESE	-	81.4 ± 1.8	0.08 ± 0.01	−24 ± 2
2d	PLGA 70:30	ESE	RhB	81.1 ± 7.9	0.07 ± 0.01	−22 ± 1
3a	Branched PLGA	NPM	-	131 ± 6.0	0.08 ± 0.01	−32 ± 2
3b	Branched PLGA	NPM	RhB	162.0 ± 9.0	0.08 ± 0.01	−33 ± 1
3c	Branched PLGA	ESE	-	97.7 ± 5.7	0.10 ± 0.01	−27 ± 2
3d	Branched PLGA	ESE	RhB	89.3 ± 8.6	0.08 ± 0.03	−24 ± 1
4a	Purasorb 5002	NPM	-	145.3 ± 0.8	0.1 ± 0.01	−24 ± 3
4b	Purasorb 5002	NPM	RhB	164.6 ± 5.5	0.1 ± 0.01	−24 ± 1

RhB-model substance Rhodamine B; PDI-polydispersity index. Data represent the mean ± standard deviation (SD) of three measurements.

**Table 2 nanomaterials-11-00749-t002:** Loading efficiency of Rhodamine B into PLGA nanospheres prepared using nanoprecipitation method (NPM) or emulsification solvent evaporation method (ESE).

Batch No.	Polymer	Preparation Method	% Drug Loading Efficiency
1b	PLGA 50:50	NPM	47 ± 4
1d	PLGA 50:50	ESE	28 ± 6
2b	PLGA 70:30	NPM	56 ± 7
2d	PLGA 70:30	ESE	29 ± 3
3b	Branched PLGA	NPM	61 ± 11
3d	Branched PLGA	ESE	34 ± 4
4b	Purasorb^®^	NPM	18 ± 5

Data represent mean ± standard deviation (SD) from three measurements.

## Data Availability

The data is available on reasonable request from the corresponding author.

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
