# Peer review of "PLGA Based Nanospheres as a Potent Macrophage-Specific Drug Delivery System"

_nanomaterials, 2021, doi:10.3390/nano11030749_

Round 1
Reviewer 1 Report
The paper entitled “PLGA based nanospheres as a potent macrophage-specific drug delivery system” by Barbora Boltnarova , Jana Kubackova , Josef Skoda , Alzbeta Stefela , Monika Smekalova , Petra Svacinova , Ivona Pavkova , Milan Dittrich , Daniel Scherman , Jarmila Zbytovska , Petr Pavek , Ondrej Holas presents a study concerning the interaction of PLGA nanoparticles with bone marrow-derived macrophages, mouse normal hepatocytes and immortalized human hepatic cell lines. Authors use various types of PLGA polymers and two types of nanoparticle preparation procedures. Synthesized PLGA nanoparticles were evaluated in terms of toxicity, cellular uptake, and inflammatory effect. The topic of the work is quite interesting and is in the scope of the Journal, however, some points have to be addressed before publication:
More details concerning the characterization of NP should be provided, in which condition the measurements were performed, in water, in a buffer? It may affect obtained values.
RhB is a water-soluble compound, did you observe leakage of RhB from synthesized PLGA NPs? How that leakage was taken into account in your experiments?
Author Response
Dear reviewer
On behalf of my co-authors, I would like to thank you for your constructive and insightful feedback on our manuscript. We found the feedback valuable and helpful in revising our manuscript and we carefully considered and addressed the provided comments. Please see our revised manuscript.
Replies to reviewers’ comments:
More details concerning the characterization of NP should be provided, in which condition the measurements were performed, in water, in a buffer? It may affect obtained values.
The material and methods section has been extensively adopted, details regarding granulometric characterization are now provided. Moreover, individual AFM topological scans for all PLGA types were added to verify spherical shape and size uniformity of prepared nanospheres. Please see sections 2.3 and 3.1. (Lines 162-184 and 303-307)
RhB is a water-soluble compound, did you observe leakage of RhB from synthesized PLGA NPs? How that leakage was taken into account in your experiments?
Release of RhB was inevitably observed during our preliminary experiments. Please see the supplementary data. The experiments were done with freshly prepared and purified nanospheres and drug loading efficiency (DLE) assessment was done at the same time point as cells treatment. DLE values that are used for cell uptake experiments results take the leakage of RhB into account. The text was adopted to clarify how the leakage was taken into consideration. (Please see lines 230-231, 175-177)
Yours sincerely,
Ondrej Holas
Reviewer 2 Report
Review of the article "PLGA based nanospheres as a potent macrophage-specific drug 2 delivery system". A large and interesting study is presented. The authors propose an easy-to-manufacture system for delivering drugs to a range of immune cells. Despite the overall positive assessment of the manuscript, I have a number of questions and suggestions for improving the text.
1. Introduction misleads readers about the results planned by the authors. From the introduction one gets the impression that the authors have achieved complete encapsulation of the dye in the polymer. It seems to me that this needs to be corrected, it is obvious that the dye comes out of the polymer. Obviously, this affects the reasoning of the authors and the manner of the story, but it is inevitable. Readers must not be misled. It has long been known that PLGA is a polymer that is actively used in the manufacture of matrices for long-term controlled release of drugs. It should be noted that for all investigated PLGA copolymers the possibility of controlled release of drugs with a molecular weight of about 1 kDa (10.1016 / j.xphs.2020.11.006; 10.1002 / app.50293; 10.1021 / acs.molpharmaceut.9b01188; 10.3390 / pharmaceutics12121165) was shown. There are record holders manufacturing PLGA-based matrices capable of separating molecules weighing about 50 kDa (10.1016 / j.reactfunctpolym.2020.104550 and 10.1080 / 09205063.2020.1760699). It should be noted that Rodomin B has a mass of about 0.5 kDa. It is known to leave the PLGA matrix perfectly (10.1016 / j.msec.2019.110191). I believe authors should clearly write about this in the introduction to the manuscript. Tell that Rodomin B comes from PLGA-based polymers. Tell that even huge protein molecules can leave PLGA-based polymers. Perhaps, according to the authors, this is not critical ... But ...
2. Measurement of the size of nanoparticles. Typically, nanoparticle sizes are presented as a size distribution and confirmed by microscopy. In this case, only extreme values ​​and a huge AFM sweep are given. It is not possible to compare the sizes. In addition, DLS measures the hydrodynamic diameter. Can the authors present the material in a more representative way?
3. When assessing the size of nanoparticles, the authors use DLS. For this, the authors must know the refractive index of nanoparticles with and without rhodamine. It is obvious that the refractive index of nanoparticles with and without a dye can differ significantly. How did the authors estimate the refractive index of nanoparticles? This should be stated in the Materials and Methods section.
4. The authors used Bone marrow-derived macrophages and two hepatocyte cell lines. It is obvious that 70% of all nanoparticles are retained in the liver, which is why hepatocytes are used as a control. However, the article does not provide information on this feature. Authors need to write! This will help readers understand the authors' intent. Another question is how good is this control? Nanoparticles in the liver are not retained by hepatocytes ...
5. If I am not mistaken, in many immune cells there is a direct relationship between the RNA levels of cytokines and the amount of cytokines themselves. Is this rule respected in Bone marrow-derived macrophages?
Author Response
Dear reviewer
On behalf of my co-authors, I would like to thank you for your very detailed, constructive and insightful feedback on our manuscript. We found the feedback valuable and helpful in revising our manuscript and we carefully considered and addressed the provided comments. Please see our revised manuscript.
Replies to reviewers’ comments:
Introduction misleads readers about the results planned by the authors. From the introduction one gets the impression that the authors have achieved complete encapsulation of the dye in the polymer. It seems to me that this needs to be corrected, it is obvious that the dye comes out of the polymer. Obviously, this affects the reasoning of the authors and the manner of the story, but it is inevitable. Readers must not be misled. It has long been known that PLGA is a polymer that is actively used in the manufacture of matrices for long-term controlled release of drugs. It should be noted that for all investigated PLGA copolymers the possibility of controlled release of drugs with a molecular weight of about 1 kDa (10.1016 / j.xphs.2020.11.006; 10.1002 / app.50293; 10.1021 / acs.molpharmaceut.9b01188; 10.3390 / pharmaceutics12121165) was shown.There are record holders manufacturing PLGA-based matrices capable of separating molecules weighing about 50 kDa (10.1016 / j.reactfunctpolym.2020.104550 and 10.1080 / 09205063.2020.1760699). It should be noted that Rodomin B has a mass of about 0.5 kDa. It is known to leave the PLGA matrix perfectly (10.1016 / j.msec.2019.110191). I believe authors should clearly write about this in the introduction to the manuscript. Tell that Rodomin B comes from PLGA-based polymers. Tell that even huge protein molecules can leave PLGA-based polymers. Perhaps, according to the authors, this is not critical ... But ...
Thank you for such a constructive comment on the manuscript. It has never been our intention to mislead potential readers. We have modified the introduction and several other parts of the manuscript to better reflect the actual PLGA usability as drug delivery systems. Please see lines 90-101). Moreover, the data regarding release of RhB from PLGA nanospheres have been added and text adopted accordingly (Please see a supplementary data and lines 230-231, 175-177).
Measurement of the size of nanoparticles. Typically, nanoparticle sizes are presented as a size distribution and confirmed by microscopy. In this case, only extreme values and a huge AFM sweep are given. It is not possible to compare the sizes. In addition, DLS measures the hydrodynamic diameter. Can the authors present the material in a more representative way?
The material and methods section has been extensively extended, details regarding granulometric characterization are now provided. Moreover, individual AFM topological scans for all PLGA types have been added to verify spherical shape and size uniformity of prepared nanospheres. Please see sections 2.3 and 3.1. (Lines 162-184 and 303-307)
When assessing the size of nanoparticles, the authors use DLS. For this, the authors must know the refractive index of nanoparticles with and without rhodamine. It is obvious that the refractive index of nanoparticles with and without a dye can differ significantly. How did the authors estimate the refractive index of nanoparticles? This should be stated in the Materials and Methods section.
We have used the intensity size distribution protocol fully aware of different RI of non-loaded and loaded nanospheres. Size distribution by intensity is not affected by differing RI as stated on manufacturers website: Refractive index values of nanoparticles - Required? (materials-talks.com). We have adopted material and methods section (Please see Lines 162-184)
The authors used Bone marrow-derived macrophages and two hepatocyte cell lines. It is obvious that 70% of all nanoparticles are retained in the liver, which is why hepatocytes are used as a control. However, the article does not provide information on this feature. Authors need to write! This will help readers understand the authors' intent. Another question is how good is this control? Nanoparticles in the liver are not retained by hepatocytes ...
Thank you for the valuable comment.
Nanoparticles are usually retained in liver resident macrophages, Kupffer cells, a part of Mononuclear Phagocytic System (MPS), but not in hepatocytes that constitute more than 80% of the liver. Therefore, we used two models. First, we used bone marrow-derived monocyte-based macrophages, which represent differentiated macrophages that can reside in specific tissues including the liver. In addition, we used hepatocyte cells. AML-12 cells represent the mouse differentiated hepatocyte model with phenotype and morphology similar to normal mouse hepatocytes and HepG2 cells represent a human hepatocyte-derived cells. These two different cell models, macrophages and hepatocytes, were used to assess whether nanoparticles are specifically delivered into macrophages, but not into hepatocytes. In macrophages, an encapsulated drug may have anti-inflammatory effects, however, in hepatocytes, it may cause adverse effects.
This was our intention. In the revised manuscript, we highlight this aim again in Introduction and Results sections (please see Lines 78-83; 342-346, 423-435).
Our data suggest that PLGA nanoparticles selectively undergo uptake by macrophages, but no significant fluorescence is detected in neither mouse nor human hepatocyte cells (Figure 2). This indicates macrophage (MPS)-specific uptake, but no hepatocyte delivery of our nanoformulations.
In addition, we show that nanoparticles taken up into macrophages do not trigger any immune activation (see point below).
If I am not mistaken, in many immune cells there is a direct relationship between the RNA levels of cytokines and the amount of cytokines themselves. Is this rule respected in Bone marrow-derived macrophages?
Thank you for the comments. Yes, we can expect a very good linear correlation and qualitative response between mRNA and protein expression, even though the relationship may not be absolute. Based on our experience with detection of cytokines with ELISA assay in macrophages, mRNA and protein expression correlate well and give us precise information if there is any immune response after delivery of nanoparticles into macrophages. In agreement, mRNA expression of cytokines is a widely accepted approach in analysis of immune activation in bonne-marrow macrophages. Last year there were several papers showing correlation between TNFα and IL1-beta mRNA and protein expression in bone-marrow derived macrophages after inflammatory stimulation (DOI: 10.1016/j.intimp.2020.106849; 10.1038/s41419-020-2590-1; 10.2147/DDDT.S243420; doi.org/10.1016/j.jep.2020.112637; 10.21037/atm.2019.11.126). Please see Lines 452-453)
Sincerely yours,
Ondrej Holas
Round 2
Reviewer 1 Report
Accept in the present form
Reviewer 2 Report
Done